# Associations between Adverse Childhood Experiences (ACEs) and Lifetime Experience of Car Crashes and Burns: A Cross-Sectional Study

**DOI:** 10.3390/ijerph192316036

**Published:** 2022-11-30

**Authors:** Kat Ford, Karen Hughes, Katie Cresswell, Nel Griffith, Mark A. Bellis

**Affiliations:** 1Public Health Collaborating Unit, School of Medical and Health Sciences, College of Human Sciences, Bangor University, Wrexham LL13 7YP, UK; 2World Health Organization Collaborating Centre on Investment for Health and Well-Being, Policy and International Health, Public Health Wales, Wrexham LL13 7YP, UK; 3Faculty of Health, Liverpool John Moores University, Liverpool L2 2ER, UK

**Keywords:** adverse childhood experiences, child maltreatment, burns, injury, prevention, road traffic accidents

## Abstract

Unintentional injury is a significant cause of disease burden and death. There are known inequalities in the experience of unintentional injuries; however, to date only a limited body of re-search has explored the relationship between exposure to adverse childhood experiences (ACEs) and unintentional injury. Using a cross-sectional sample of the adult general public (*n* = 4783) in Wales (national) and England (Bolton Local Authority), we identify relationships between ACE exposure and experience of car crashes and burns requiring medical attention across the life course. Individuals who had experienced 4+ ACEs were at significantly increased odds of having ever had each outcome measured. Furthermore, compared to those with no ACEs, those with 4+ were around two times more likely to report having had multiple (i.e., 2+) car crashes and over four times more likely to report having had burns multiple times. Findings expand the evidence base for the association between ACEs and negative health consequences and emphasise the need for effective interventions to prevent ACEs and their impact on life course health and well-being. Such knowledge can also be used to develop a multifaceted approach to injury prevention.

## 1. Introduction

Unintentional injury is a significant cause of disease burden and death. Globally, an estimated 3.16 million unintentional injury-related deaths are recorded each year, around a third of which result from road traffic collisions [1]. Millions more individuals experience non-fatal injury; creating a significant public health problem. Unintentional injury places a burden upon health services through short- and long-term treatment and rehabilitation costs. However, unintentional injury at any stage during the life course can have lasting consequences for the individuals affected and cause long-term physical and mental illness and disability. In the UK, unintentional injury accounts for only a small proportion of all-cause mortality yet leads to a high burden of years of life lost, accounting for 826,455 disability adjusted life years (DALYs) in 2019 [2]. However, the majority of unintentional injuries are preventable. There are known inequalities in the experience of unintentional injuries, although this can vary by injury type. For example, males are at greater risk of road traffic injuries [3] and head injury has been found to be more common in those residing in more deprived areas [4]. It is important that the risk factors and determinants for unintentional injury are fully understood to ensure effective preventative action.

A growing body of evidence links exposure to adverse childhood experiences (ACEs) such as child maltreatment, witnessing violence in the home and parental substance misuse or mental illness with a range of poor life course outcomes. ACEs such as child maltreatment can directly cause physical injury or childhood death [5,6], with a high prevalence of ACEs having been identified in samples of children who have died [6,7]. However, exposure to ACEs can also impact children’s neurological, social and emotional development with implications for behaviour and health throughout life [8]. Studies have found strong relationships between exposure to multiple ACEs and increased propensity for risk-taking and health-harming behaviours such as alcohol and drug use [8,9], which can increase vulnerability to unintentional injury. Numerous studies have also shown strong associations between ACEs and increased risk of intentional injuries, including violence and self-directed injury such as self-harm and suicide attempt [8,10]. ACEs are also associated with increased use of healthcare services, including attendance at emergency departments or overnight hospital stays [11,12]. Although, many relationships between ACEs and poor health outcomes are well established, the relationships between ACEs and markers of unintentional injury remain underexplored. 

A limited range of research has explored associations between ACEs and unintentional injury in general population samples. In 1998, the foremost study into ACEs found that adults with 4+ ACEs were 1.6 times more likely to have ever had a skeletal fracture (used as a proxy for unintentional injuries) than those with no ACEs [13]. However, since this publication most research on ACEs and unintentional injury has focused on brain injury [14]. A US study found that adults with 4+ ACEs were over three and a half times more likely to have experienced an acquired brain injury than those with no ACEs. Associations were also found between brain injury and specific ACE types, including sexual, emotional and physical abuse, and household mental illness and substance abuse [15]. Increased risk of brain injury has also been found in adolescents exposed to 4+ ACEs [16], and in samples of vulnerable individuals such as those incarcerated, homeless, at high risk of psychosis and with severe mental illness [17]. 

For other markers of unintentional injury, a UK study found adults reporting 4+ ACEs were over two times more likely to report having ever broken a bone than those with no ACEs [18] while in the US ACEs have been associated with serious injury. A nationally representative sample of women identified that those who had been exposed to physical or sexual abuse in childhood were at increased risk of sustaining a serious injury (spinal cord, neck or head) in adulthood compared to those who had not experienced these types of abuse [19]. Further, in young adults, specific ACEs have been associated with past year serious injury (physical and sexual abuse, emotional neglect) and being in a motor vehicle collision (physical abuse, emotional neglect) [20]. Other research on ACEs and unintentional injury has predominately been conducted within populations affected by injury [21,22]. For example, a US study of burn patients identified a higher prevalence of ACEs (17.0% reporting 4+) compared to the general population (14.5%) [23]. 

Few studies explore associations between ACEs and types of injury commonly described as unintentional in general populations outside of the US. Therefore, we used a cross-sectional sample of the adult general public within Wales (national) and England (Bolton Local Authority) to explore relationships between ACEs and experience of life course markers of two unintentional injury types: involvement in car crashes and burns. Better understanding of risk factors for unintentional injury can assist in the development of appropriate prevention efforts. 

## 2. Materials and Methods

Between December 2020 and March 2021, a telephone survey was conducted with residents aged 18 years and over in Wales (national) and Bolton Local Authority. Data collection was piloted on the 15, 16 December 2020, with main study data collection between January–March 2021. During this time Wales and England experienced a period of heightened COVID-19 restrictions; limits were placed on household mixing, social interaction, hospitality, and some non-essential retail, and the use of face coverings in indoor public places were required. 

Using ACE prevalence from other UK ACE studies [24,25], a target sample of 4000 individuals was set to ensure a minimum of 800 respondents in the 4+ ACEs category. A professional market research company (MRC) undertook data collection using a random stratified sampling approach. Landline and mobile telephone contacts (obtained from a commercial provider) were stratified by Health Board area in Wales only, then by residential deprivation based on rankings in the relevant Indexes of Multiple Deprivation (IMD; [26,27]) and age. The IMD is widely used to classify the socioeconomic status of UK areas, however, there is variation in indices across the UK nations. Following difficulty accessing younger age participants using telephone sampling, an online version of the survey was disseminated by the MRC through a commercial provider to an online panel sample (individuals paid to take part in online research). Additionally, in Bolton, targeted community sampling within a local college and local agencies, was undertaken (with potential participants directed to the online survey). 

The study inclusion criteria were: resident in the study areas, aged 18 years or over and cognitively able to participate. All potential participants were provided with a description of the study (verbal or electronic) including its purpose and voluntary, anonymous and confidential nature. It was also made clear that a decision to not participate or to withdraw from the study would not affect their rights, future health treatment or service provision and that participants did not have to answer all questions. Informed consent (opt-in) was recorded for all participants either verbally or electronically. Following survey completion, participants were directed to a web-link providing contact details for the research team and appropriate support services. Study materials were available in English, with Welsh language versions available in Wales and other language versions available on request. Telephone calls were undertaken between 9 a.m.–9 p.m. Monday-Friday and 10 a.m.–4 p.m. Saturday-Sunday. The survey took on average 20 min to complete.

Telephone contact was made with 12,536 individuals, of whom 1.8% (*n* = 230) did not meet the study inclusion criteria and 63.5% (*n* = 7964) declined to participate. 3984 completed the questionnaire and 358 did not meet the age quota in their area. Thus, the telephone participation rate was 32.4% (3984/12,306) of all eligible telephone participants or 33.3% (3984/11,948) of eligible individuals meeting the quota sampling. A participation rate for the online sample could not be calculated. Individuals who did not answer demographic questions, could not be allocated an ACE count or did not respond to outcomes of interest were removed, resulting in a sample of 4783 for analysis. 

Childhood exposure to ACEs (before the age of 18; see Appendix A for outcome questions and response options) were measured using an adapted version of the Centers for Disease Control and Prevention short ACEs survey tool [28]. The nine ACE types included physical, verbal, or sexual abuse; exposure to domestic violence; parental separation; household member incarceration, mental illness, and alcohol or substance abuse. Consistent with international literature [5], ACE exposure was categorised to a count: 0 ACE, 1 ACE, 2–3 ACEs, 4+ ACEs. Experience of life course injury was measured by asking participants how many times in their lives they had been in a car crash (regardless of whose fault it was) and had a burn requiring medical attention (response options: never, once, 2–5 times, 6–10 times, more than 10 times; see Appendix A). Due to small numbers across frequency categories, outcomes were coded to never, ever; with a separate variable created to explore an incremental effect—coded never, once and multiple (i.e., 2+) times. Respondents’ gender (male; female; other), age (five-year groups), and ethnicity (self-reported UK census categories) were also collected. For the purposes of analysis, age was categorised into ten-year groupings (18–29; 30–39; 40–49; 50–59; 60–69; 70+) and due to low levels in ethnic minority groups, ethnicity was re-categorised (white, other). The MRC converted postcode of residence into the deprivation quintile for the Lower Super Output Area (LSOA) using the respective IMD of the study area (1 = most deprived to 5 = least deprived). 

Statistical analysis was performed using SPSS v25 (IBM, Armonk, NY, USA). Bivariate analysis used chi squared, with binary logistic regression analysis (enter method) used to measure independent associations between ACEs and having ever had the injury outcomes, controlling for participant socio-demographics (age, gender, ethnicity, deprivation), study area (Bolton or Wales) and survey method (telephone or online). Multinomial logistic regression analysis (enter method) including the same confounders was run to explore the differences amongst those experiencing injury outcomes once or multiple times compared to never. Additional multinomial logistic regression analysis was also run to explore associations between the injury outcomes and all individual ACE types measured (see Appendix A). 

## 3. Results

Table 1 shows the sample demographics. Around six in ten respondents were female (62.2%) and a similar proportion (63.3%) aged 50 and over, with 93.7% of the sample reporting white ethnicity. Proportions in each deprivation quintile ranged from 17.7% (quintile 3) to 25.2% (quintile 1, most deprived). Half of the sample (49.6%) reported experiencing at least one ACE before the age of 18, with 10.5% reporting exposure to 4+ ACEs. Higher ACE counts were more common in younger participants, those from more deprived communities, those participating online and those from Wales.

### 3.1. Car Crash

Just over half of the sample (52.6%) reported having ever been in a car crash (24.4% once, 28.2% multiple times; see Appendix A). The proportion having ever been in a car crash increased with ACE count from 48.5% of those with 0 ACEs to 62.5% in those with 4+ ACEs (see Table 2). In logistic regression analysis, having ever been in a car crash remained independently associated with higher ACEs (4+ ACEs, AOR 1.9; Table 2). 

A higher proportion of individuals with multiple ACEs reported having been in multiple car crashes (Appendix A). Thus, the proportion having been in a car crash once increased from 23.4% with 0 ACEs to 25.5% with 4+ ACEs, while the proportion having been in multiple car crashes increased from 25.2% to 37.1%, respectively. In multinomial logistic regression (Table 3), compared with those with no ACEs, respondents with 4+ ACEs were 1.5 times and 2.3 times more likely to have been in a car crash once or multiple times, respectively. Odds of having been in a car crash once were also increased in those with 2–3 ACEs, whilst odds of having been in multiple car crashes were increased in those with 1 or 2–3 ACEs (Table 3). Males, those aged 40–49 and those living in more deprived areas were more likely to report having been in a car crash (once and multiple times), whilst residing in England was associated with increased odds of having been in multiple car crashes (see Appendix A). 

A separate multinomial model explored relationships between individual ACE types and car crashes. Parental separation or divorce was associated with increased odds of having been in a car crash once, whilst physical, sexual or emotional abuse or household member incarceration was associated with increased odds of having been in multiple car crashes (see Appendix A). 

### 3.2. Burns

One in ten adults (11.6%) reported having ever had a burn that required medical attention (9.6% once, 2.1% multiple times). The proportion having ever had a burn increased with ACE count from 9.0% of those with 0 ACEs to 20.3% in those with 4+ ACEs (Table 2). In logistic regression analysis, having ever had a burn remained independently associated with higher ACEs (4+ ACEs, AOR 2.2; Table 2). 

A higher proportion of individuals with ACEs reported having had a burn multiple times (Appendix A). Thus, the proportion having a burn once increased from 7.9% with 0 ACEs to 14.9% with 4+ ACEs, while the proportion having had multiple burns increased from 1.1% to 5.4%, respectively. In multinomial logistic regression (Table 3), compared with those with no ACEs, respondents with 4+ ACEs were 2.0 and 4.1 times more likely to have had a burn once or multiple times, respectively. Odds of having had a burn once or multiple times were also increased in those with 2–3 ACEs. There was no significant increase in odds associated with reporting 1 ACE. Males, those aged 70+ and those living in more deprived areas were more likely to report having had a burn (once or multiple times), whilst completing the survey online was associated with increased odds of having had multiple burns (see Appendix A).

A separate multinomial model explored relationships between individual ACE types and burns. Household member incarceration was associated with increased odds of having had a burn once or multiple times. Household member mental illness was associated with increased odds of having had one burn and sexual abuse was associated with increased odds of having had multiple burns (Appendix A). 

## 4. Discussion

Using a UK general population sample, this study has identified relationships between exposure to ACEs and lifetime experience of car crashes and burns; two major markers of unintentional injury. Moreover, our study finds that ACEs are associated with experiencing these injury outcomes multiple times. In line with previous ACE research, we found that odds of experiencing car crashes and burns both ever and multiple times increased with the number of ACEs individuals reported in their childhood histories. Further, these relationships were independent of other known risk factors for injury, including social deprivation [29]. Thus findings expand the evidence base for the cumulative association between ACEs and negative health consequences and identify ACEs as an important consideration for injury prevention. 

There are various mechanisms by which ACEs may be linked to involvement in car crashes or injuries such as burns. During childhood, children’s safety depends on the protection they receive from their caregivers and injury risk can occur where such protection is compromised by ACEs such as caregiver substance misuse, mental illness, conflict and involvement in criminal behaviour. However, exposure to ACEs can also impact individuals’ neurological functioning, health and well-being across the life course. Thus, studies suggest that child maltreatment can impact executive functioning, emotional regulation and threat response [30,31] which might increase vulnerability to unintentional injury; with psychological functioning, impulsivity and mental health disorders being significant risk factors for injury [20]. There is also substantial literature linking ACEs to health-harming behaviours (e.g., drug and alcohol misuse; [32]) which may impair judgement, decrease attention to the environment and place individuals at higher risk of injury, particularly for motor vehicle collisions and interpersonal violence [33,34,35,36]. It is possible that these behavioural risk factors and the impact of exposure to ACEs on executive functioning may help to explain why individuals with ACEs have higher odds of injury. 

We found an increased prevalence of car crashes was associated with multiple ACE exposure, with respondents with 4+ ACEs being over two times more likely to have been in multiple car crashes compared with those with no ACEs. While we did not specifically ask about injury resulting from car crashes, findings provide an indication of increased vulnerability to this major cause of unintentional injury in individuals exposed to ACEs. In the UK there has been major progress in reducing the number of road deaths and casualties, and the 2019 UK Road Safety action plan highlights actions being taken to address risk factors for such injury [37]. However, road traffic crashes remain a key cause of unintentional injury and injury-related mortality. As such, in 2020 the United Nations (UN) General Assembly adopted a new resolution on road safety (A/RES/74/299; [38]). Further, in conjunction with the World Health Organization, the UN General Assembly prioritize a holistic approach to improving road safety through their Global Plan for the Decade of Action for Road Safety 2021–2030 [39]. Due to the gender differentials identified in road injury and risk-taking (with males at increased risk), the plan advocates a gender perspective in transport planning and a recognition that road safety cuts across policy agendas including child health and gender. However, the global plan does not include an exploration of the other possible underlying risk factors for motor vehicle collisions. Our findings suggest that childhood adversity may be an important risk factor. As such, it is important that policymakers understand the role that ACEs may play in the experience of this type of unintentional injury. 

In line with previous research, we found higher levels of burns reported by individuals of male gender [40]. However, even when adjusting for demographic confounders, individuals with 4+ ACEs were over two times more likely to report having ever had a burn and over four times more likely to report having had a burn multiple times. Such findings align with other research indicating that individuals who have experienced ACEs are at increased risk of burns [21,23]. ACEs have also been associated with increased risk of a more complicated recovery following a burn, with burn patients with higher ACE counts having been found to be less resilient and more likely to have depression and post traumatic stress disorder (PTSD) symptoms [23]. Thus researchers have indicated that ACE screening may help detect burn patients at risk of poor outcomes [23]. To our knowledge no studies of screening or routine enquiry have been conducted in population groups who have suffered injury to explore patient acceptability, feasibility or other outcomes following ACE enquiry. Previous pilots of routine enquiry in health settings (general practice and health visiting services) in the UK have found support for the feasibility and acceptability of enquiry amongst service users and practitioners, yet the long-term implications of enquiry for service users and service demand are unknown and the overall evidence base for routine enquiry is limited [41]. Although for children, injury presentations may raise concerns about childhood abuse, for adults, the experience of ACEs as a factor in their injury may not be considered. 

In the UK, guidance for public health and health professionals recognises that ACE exposure increases the risk of poor health outcomes and health-harming behaviours across the life course [42]. Despite this, as the relationship between ACEs and unintentional injury beyond traumatic brain injury is relatively underexplored, an understanding of these relationships and their impact on poor outcomes has not been communicated to health professionals or adapted into policy. However, the role that the experience of ACEs may have in injury needs to be acknowledged. As the majority of unintentional injuries are preventable it is important that the risk factors and determinants for unintentional injury are fully understood and communicated to ensure effective preventative action and responses are taken. Given the staggering societal costs associated with both ACEs [43] and unintentional injury [44] it is important that new opportunities for the delivery of injury prevention interventions are identified. 

There are a number of study limitations which should be acknowledged in interpretation of these findings. Compliance could not be measured for those completing the survey online and although similar to other telephone surveys [45], compliance for telephone participation was low (33.3%). Furthermore, we did not record reasons for declining to participate and therefore we are unable to identify any non-participation bias. All measures were self-reported and subject to reporting bias, accurate reporting, and recall. Our measurement of ACEs was restricted to nine commonly measured ACE types; yet there are a wide range of other adversities that participants may have been exposed to during childhood, such as neglect, community violence and displacement. Further, the use of an ACE count measure does not account for the timing nor extent of ACE exposure during childhood. Our measurement of unintentional injuries covered the life course, so we are unable to identify whether injuries occurred in childhood or adulthood. Equally, while we focused on car crashes and burns as markers of unintentional injury, we cannot rule out that some such experiences, whether in adulthood or childhood, may have been intentional. In analysis of individual ACEs, physical, sexual and emotional abuse were associated with involvement in multiple (but not single) car crashes, as was parental incarceration. However, there was no relationship between physical abuse or emotional abuse and burns; with multiple burns associated with sexual abuse and parental incarceration. We did not measure other types of unintentional injury (e.g., falls, near drowning, poisoning) which may be of interest for future research. Our measure of deprivation used IMD quintile, based on current residential postcode, and we did not measure childhood deprivation. Further, our sample was predominantly of white ethnicity (94%; vs. 95% in the Welsh national population and 82% in Bolton) and consequently under-represented minority ethnic groups. Further research is required to examine ACEs and their relationships with injuries in more diverse population groups. To account for cultural differences that may exist across study locations (Wales, England) and bias introduced through survey method, these cofounders were adjusted for in our analyses. Finally, the study was cross-sectional and therefore we cannot infer causality. Examining these relationships in prospective, longitudinal samples would help to explain these relationships further. 

## 5. Conclusions

Exposure to ACEs is associated with increased odds of having been involved in car crashes and experiencing severe burns across the life course. Findings highlight the need for effective interventions to prevent ACEs and reduce their impacts on health and well-being. A better understanding of the relationships between ACEs and unintentional injury, and the mechanisms that link childhood adversity to injury risks, can benefit the development of multifaceted approaches to injury prevention.

## Figures and Tables

**Table 1 ijerph-19-16036-t001:** Sample demographics and distribution of ACE count.

	Sample		ACE Count					
			0	1	2–3	4+		
		n	%	%	%	%	X^2^	*p*
*All*		4783	50.4	22.1	17.0	10.5		
*Gender*	Male	1809	51.8	21.6	17.2	9.3		
	Female	2974	49.6	22.4	16.8	11.2	5.157	0.161
*Age group*	18–29	448	34.4	24.1	23.2	18.3		
	30–39	521	39.5	21.5	21.7	17.3		
	40–49	787	45.7	20.6	21.0	12.7		
	50–59	1037	47.7	22.6	18.1	11.6		
	60–69	898	54.0	22.9	14.9	8.1		
	70+	1092	65.2	21.4	10.0	3.4	261.833	<0.001
*Ethnicity*	White	4483	50.3	22.2	17.0	10.5		
	Other	300	52.7	19.7	17.3	10.3	1.190	0.755
*Deprivation*	1—most	1204	45.3	21.3	19.7	13.7		
*quintile*	2	865	48.0	21.8	18.8	11.3		
	3	847	50.2	23.8	17.0	9.0		
	4	926	54.2	21.3	15.1	9.4		
	5—least	941	55.7	22.5	13.7	8.1	52.260	<0.001
*Survey*	Telephone	3944	52.0	21.9	16.6	9.5		
*method*	Online	839	43.3	22.8	18.7	15.3	34.302	<0.001
*Study area*	Bolton	1865	50.9	22.9	17.3	8.8		
	Wales	2918	50.1	21.5	16.8	11.5	9.276	0.026

**Table 2 ijerph-19-16036-t002:** Proportions ever reporting injury outcomes and adjusted odds ratios, by ACE count.

		Car Crash (Ever)				Burn (Ever)			
		% #	AOR	95% CIs	*p*	% #	AOR	95% CIs	*p*
	All	52.6				11.6			
ACE count	0	48.5	Ref.		<0.001	9.0	Ref.		<0.001
	1	52.9	1.22	1.05–1.41	0.011	11.3	1.22	0.96–1.55	0.101
	2–3	58.3	1.49	1.26–1.77	<0.001	14.5	1.54	1.21–1.97	0.001
	4+	62.5	1.88	1.53–2.32	<0.001	20.3	2.24	1.71–2.93	<0.001

AOR = adjusted odds ratio; CI = confidence interval (95%); Ref = reference category. Logistic regression models controlled for gender, age group, ethnicity, deprivation quintile, survey method and study area. Proportions and adjusted odds ratios for all variables are presented in Appendix A. # Bivariate analysis (chi squared) *p* < 0.001.

**Table 3 ijerph-19-16036-t003:** Multinomial logistic regression analysis: relationship between ACE count and frequency of injury outcomes.

		Car Crash							Burn						
			Once			2+ Times				Once			2+ Times		
		Reference Category *	AOR	95% CIs	*p*	AOR	95% CIs	*p*	Reference Category *	AOR	95% CIs	*p*	AOR	95% CIs	*p*
ACE count	0	<0.001	Ref.			Ref.			<0.001	Ref.			Ref.		
	1		1.15	0.96–1.38	0.130	1.28	1.07–1.54	0.007		1.15	0.89–1.49	0.292	1.74	0.98–3.08	0.060
	2–3		1.43	1.17–1.75	0.001	1.54	1.26–1.89	<0.001		1.45	1.11–1.89	0.006	2.17	1.22–3.85	0.008
	4+		1.53	1.18–1.97	0.001	2.30	1.80–2.93	<0.001		1.95	1.45–2.62	<0.001	4.13	2.34–7.29	<0.001

AOR = adjusted odds ratio; CI = confidence interval (95%); Ref = reference category. * Reference category for dependent variables was none. Multinomial logistic regression models controlled for gender, age group, ethnicity, deprivation quintile, survey method and study area. Adjusted odds ratios for all variables are presented in Appendix A.

## Data Availability

The data presented in this study are available on request from the corresponding author.

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
