# Peer review of "Associations between Adverse Childhood Experiences (ACEs) and Lifetime Experience of Car Crashes and Burns: A Cross-Sectional Study"

_ijerph, 2022, doi:10.3390/ijerph192316036_

Round 1
Reviewer 1 Report
This study examined associations between adverse childhood experiences (ACEs) and unintentional injury among a sample of adults in England and Wales. The authors used a standard ACEs questionnaire to assess childhood experiences and investigated associations with participants’ history of experiencing 4 outcomes at any point in their life: car crashes, burns, broken bones, and requiring stitches. The authors found that individuals who reported experiencing more ACEs faced an increased odds of each measure of unintentional injury, and they were also more likely to experience these outcomes multiple times in their life. The manuscript is well-written and the findings are clearly communicated. A strength of this study is the focus on an understudied outcome in the ACEs literature and the sample, which was not only sizeable, but recruited for the purpose of studying ACEs (and therefore, was sufficiently powered). There are a few issues that I identified that would warrant further attention:
1) A major limitation of this paper is the lack of information regarding the timing of the outcome. While I know it is a cross-sectional study, the assessment of outcomes ever in one’s lifetime means that these health issues could have even occurred prior to the participants’ ACE exposure. Of a greater concern, however, is the point the authors highlight about some of these outcomes potentially being related to the ACE exposure itself (e.g., burns, broken bones, stitches resulting from abuse or violence). As I see it, the inability to parse that out is a potentially fatal methodological flaw in this paper.
To that end, I believe this study would be strengthened substantially by the addition of sensitivity analyses to confront this issue head on and hopefully mitigate concerns regarding the validity of the results. For example, one approach could be to examine associations between individual ACEs and the outcomes in question to determine whether associations with potentially violence-related outcomes are noted with ACEs that are not violence-related (e.g., parental separation, household member incarceration). Although this may also present challenges since the abuse categories, domestic violence, and living with a family member with mental illness and alcohol or substance abuse could all be related to violence exposure, it would likely give us greater insight into whether these outcomes could themselves represent ACES.
I urge the authors to carefully think through other potential strategies to address this issue, choose what suits their study and the available data best, and provide a strong justification in both their response letter and revised manuscript.
2) Although the issue I raise above is referenced in the discussion, I think it’s a much larger methodological issue than currently described, to the point where it may even be preferable to exclude the results from those outcomes altogether. I do believe that the findings with car crashes are interesting, particularly in light of the points raised in the discussion regarding executive functioning, emotion regulation, and threat response. As a result, I think a paper focused exclusively on car crashes could be compelling in and of itself. However, if the authors prefer to keep the more methodologically questionable outcomes in their analysis, there also needs to be more careful attention to describing the plausible mechanisms linking these ACE exposures and burns, broken bones, and requiring stitches in the discussion.
3) With regard to the presentation of findings: the results tables that are provided are quite busy, and in some cases, unnecessary. Since the focus of the study is on the relationship between ACEs and unintentional injuries, it is unclear why the authors provided the prevalence of the outcomes they studied by all study variables in Table 1. To make this a bit more digestible for the reader, I would suggest presenting standard demographics in Table 1 (e.g., n’s and %/means for all the study variables) and then present differences in the distribution of ACEs by study variables (meaning, have the columns present the prevalence of the exposure rather than the outcome). The percentages that are currently provided in this table are already described in the results section, so I don’t see it as necessary to have all of this information duplicated.
4) Similarly, it is unclear to me why all the study covariates are presented in Tables 2 and 3, and why there are lines demarcating each. I would recommend simply presenting associations with ACEs in these tables and removing the other results to guard against potential biased interpretation of findings due to the Table 2 fallacy (for more information, see: https://academic.oup.com/aje/article/177/4/292/147738)
Reviewer 2 Report
The authors present an interesting and important topic for consideration in this study. I have some smaller revision suggestions to improve readability and expand upon the conclusions; however, I also have one major concern about the validity of the conclusions drawn from these results.
Minor revisions
Please proofread the article again to catch any remaining typos. There were some places where there were missing words and/or letters, or incorrect or confusing punctuation was used.
The authors should provide an explanation or justification (possibly in the Materials and Methods section) for not including the neglect dimension of ACEs in their survey.
The tables are challenging to read. They should be reformatted so that numbers do not break across multiple lines.
The authors do not devote much space in the paper to the analysis looking at specific types of ACEs and accidental injury. They should consider dropping this analysis from the paper as it does not necessarily add anything substantial and detracts from the main findings around total ACEs and accidental injury.
There are a few places in the discussion where the authors discuss increased risk. Please update these to increased odds, as the prevalence of the outcome is to high to be able to accurately interpret the odds ratio as a risk ratio.
The authors discuss concerns of the representativeness of the sample in the limitations section; however, they do not devote any space here to a discussion of the lack of racial/ethnic diversity in the sample. This should be added to and discussed in the limitations.
Major revisions
The conceptual framework presented in this paper has ACEs in childhood leading to accidental injury, presumably in adulthood or at least after the ACEs have happened. They further hypothesize that this may happen through increased risk behaviors, limited awareness of surroundings, or decreased cognitive functioning. However, as the authors acknowledge in the limitations section, the survey asked about accidental injury throughout the entire life-course. While it is good that this limitation is acknowledged, it does call into question the interpretation of the results. Overall, if the injuries happen concurrent to the ACEs, they may simply be a sign that ACEs are occurring. For a more specific example, a car crash that happened when the respondent was a child – and therefore could not have been driving the car – is very different than a car crash that happened when the respondent was an adult and potentially could have been the one driving the car.
The inability to distinguish between ACEs and accidental injury happening sequentially versus concurrently means the conclusion the authors draw may not be valid. The paper needs to be reframed and the introduction and discussion rewritten to address this.
Round 2
Reviewer 2 Report
The edits the authors have made have much improved the manuscript.